

# The effect of atmospherically relevant aminium salts on water uptake

Noora Hyttinen

Department of Chemistry, Nanoscience Center, University of Jyväskylä, FI-40014 Jyväskylä, Finland

**Correspondence:** Noora Hyttinen (noora.x.hyttinen@jyu.fi)

**Abstract.** Atmospheric new particle formation is initiated by clustering of gaseous precursors, such as small acids and bases. The hygroscopic properties of those precursors therefore affect the hygroscopic properties of aerosol particles. In this work, the water uptake of different salts consisting of atmospheric small acids and amines was studied computationally using the conductor-like screening model for real solvents (COSMO-RS). This method allows for the prediction of water activities in

atmospherically relevant salts that have not been included in other thermodynamics models. Water activities are reported here for binary aqueous salt solutions, as well as ternary solutions containing proxies for organic aerosol constituents. The order of the studied cation species regarding water activities is similar in sulfate, iodate and methylsulfonate, as well as in bisulfate and nitrate. Predicted water uptake strengths (in mole fraction) follow the orders: tertiary > secondary > primary amines, and guanidinos > amino acids. The addition of water soluble organic to the studied salts generally leads to weaker water uptake

compared to pure salts. On the other hand, water-insoluble organic likely phase separates with aqueous salt solutions, leading to minimal effects on water uptake.

## 1 Introduction

Water uptake of atmospheric aerosol has a large effect on the radiative forcing though, e.g., cloud formation. Theoretical and experimental evidence show that even a small amount of salt has a dominant effect on cloud condensation nucleus (CCN)

activity (Bilde and Svenningsson, 2004; King et al., 2012). These salts may originate from sea spray or from the nucleation and condensation of acidic and basic compounds, such as sulfuric acid and amines.

The nucleation of gas-phase acids and bases has been investigated computationally during the recent years by computing formation energies of clusters that contain various small molecules (Elm, 2021). Computational studies have shown that various amines can form molecular clusters with acidic species, such as sulfuric acid (Elm et al., 2016; Myllys et al., 2018; Kubečka

et al., 2023). Liu et al. (2022) found that guanidino-containing compounds (guanidine, agmatine) enhance methylsulfonic acid driven nucleation. Chen et al. (2022) simulated the nucleation of nitric acid with various alkylamines, alkanolamines and polyamines. They found that alkanolamines and polyamines can nucleate with nitric acid at room temperature, while ammonia and alkylamines require lower temperatures.

The activity of water in aerosol particles describes the response of the particles to variations in RH. Some experimental

studies have investigated water activities of ammonium and aminium sulfate and nitrate salts (Bonner, 1981; Clegg et al., 2013;



Sauerwein et al., 2015; Rovelli et al., 2017). For example, Rovelli et al. (2017) found that water activity in alkylaminium sulfate particles increases with the increasing alkyl group length and number of alkyl groups. However, water activity measurements of salt solutions comprising larger amines relevant salt solutions are still lacking.

This study investigates the effect of different amine and acid species involved in the atmospheric nucleation on water activity computationally. Many thermodynamics models commonly used for activity coefficient calculations, such as AIOMFAC (Zuend et al., 2008, 2011), are parametrized for only a few atmospheric ions. On the contrary, the quantum chemistry based conductor-like screening model for real solvents (COSMO-RS; Klamt, 1995; Klamt et al., 1998; Eckert and Klamt, 2002) can be used to compute thermodynamic properties of any atmospherically relevant species. For this reason, COSMO-RS was used here to predict water activities of the salt solutions of interest.

The acids selected for this study are abundant in the atmosphere. Sulfate ($SO_4^{2-}$) and bisulfate ($HSO_4^-$) are derived from anthropogenic emissions of $SO_2$, nitrate ($NO_3^-$) is produced from both anthropogenic and natural processes, and iodate and methylsulfonate ($IO_3^-$ and $CH_3SO_3^-$, respectively) are more abundant in marine environments. The following atmospherically abundant alkylamines, guanidinos and amino acids (Di Filippo et al., 2014) were selected as the bases for the calculations.

- Alkylamines: ammonia (AM), monomethylamine (MMA), dimethylamine (DMA), trimethylamine (TMA) diethyl amine (DEA), ethylenediamine (EDA) and tetramethylethylenediamine (TMEDA)

- Guanidinos: guanidine (GUA) agmatine (AGM) and arginine (ARG; also an amino acid)

- Amino acids: alanine (ALA), aspartic acid (ASP), histidine (HIS), glutamic acid (GLU), glycine (GLY) and serine (SER)

These amines have many biogenic (vegetation, biomass burning) and anthropogenic (pesticides, combustion, livestock) sources (Ge et al., 2011). Additionally, the effect of the salts on water uptake was investigated in the presence of a hydrophilic and a hydrophobic organic.

## 2 Computational methods

The COSMO-RS model (Klamt, 1995; Klamt et al., 1998; Eckert and Klamt, 2002), implemented in the BIOVIA COSMO*therm* program (BIOVIA COSMO*therm*, 2021a) (abbreviated COSMO*therm*), was used for activity coefficient and liquid-liquid equilibrium (LLE) calculations. Activity coefficient $\gamma$ of compound $i$ is computed in COSMO*therm* using the following equation:

$$\ln \gamma_i(\mathbf{x}) = \frac{\mu_i^*(\mathbf{x}) - \mu_i^{*,\circ}(\mathbf{x}^\circ, T, P)}{RT} \tag{1}$$

where $\mathbf{x}$ is the mixing state in mole fraction, $T$ is the temperature (here 298.15 K), $R$ the gas constant (in kJ K$^{-1}$mol$^{-1}$, when $\mu^*$ is given in kJ mol$^{-1}$) and $P = 10^5$ Pa reference pressure. Pseudo-chemical potential (Ben-Naim, 1987) $\mu^*$ is an auxiliary quantity defined using the chemical potential at the reference state $\mu^\circ$:

$$\mu_i^*(\mathbf{x}) = \mu_i^\circ(\mathbf{x}^\circ, T, P) + RT \ln \gamma_i(\mathbf{x}) \tag{2}$$



The pure compound ($x_i = 1$) is used as the reference state composition $\mathbf{x}^\circ$. The use of pseudo-chemical potential allows for the calculation of activity coefficients in infinitely diluted states ($x_i \to 0$).

The occurrence of liquid-liquid or solid-liquid phase separation (LLPS or SLPS, respectively) is not considered in COSMO*therm* activity coefficient calculations. Instead, the mixtures are assumed to be homogeneous. Small aerosol particles are less likely to phase separate than larger droplets (Freedman, 2020). One-phase states are therefore possible in small particles even if the

bulk solution would phase separate.

The solubility of the hydrophobic organic in the salt solutions was found using the LLE condition:

$$\gamma_i(\mathbf{x}^\alpha)x_i^\alpha = \gamma_i(\mathbf{x}^\beta)x_i^\beta \tag{3}$$

Here, $\alpha$ and $\beta$ denote the aqueous phase and the organic-rich phase. In the LLE calculation, the salt is assumed to be fully dissolved into the liquid phases.

## 65 2.1 Electrolytes in COSMO*therm*

The framework of COSMO*therm* calculations treats electrolytes (i.e., solvent and one salt) as ternary solutions (solvent, cation, anion). This affects the mole fractions (input) and activity coefficients (output) of the program. All mole fractions presented with the results were converted to the laboratory-binary framework (solvent, salt) for easier comparison with experiments.

The highest level of theory parametrization in COSMO*therm* (BP_TZVPD_FINE_21; abbreviated FINE) works poorly with

the strong charges of small ions, such as atomic ions and strong small semi-spherical ions (such as $SO_4^{2-}$, $NH_4^+$). The use of an electrolyte parametrization (BP_TZVP_ELYTE_21; abbreviated ELYTE) is recommended for up to 6 molal solutions, corresponding to around 0.1 mole fraction of salt in water. However, the use of ELYTE is not recommended for large ions. (BIOVIA COSMO*therm*, 2021b)

The different COSMO*therm* parametrizations are compared in Sect. S1 of the Supplement. Overall, the experimental water

activities agree equally well with both ELYTE and FINE parametrizations with the exception of ammonium sulfate, in which FINE significantly overestimates the experiments (see Figs S1–S3 of the Supplement). For some cations (e.g., ARG), ELYTE results are highly inconsistent with the results of FINE and the BP_TZVP_21 parametrization (abbreviated TZVP), which for most salts give fairly similar estimates. The FINE parametrization is therefore used for all of the studied salts, with the exception of ammonium sulfate, which is calculated using the ELYTE parametrization.

## 80 2.2 Input files for COSMO*therm* calculations

The input files for COSMO*therm* calculations (cosmo-files) were obtained through a series of density functional theory calculations with increasing level of theory. The process has been discussed in more detail in a previous publication (Hyttinen et al., 2020). In short, all conformers were found using the systematic conformer search algorithm in the Spartan20 program (Wavefunction Inc., 2020). The geometries of all conformers were optimized and duplicate conformers were removed using the

BIOVIA COSMO*conf* program (BIOVIA COSMO*conf*, 2021), which uses the Turbomole program (TURBOMOLE, 2020)





for the quantum chemistry calculations. The final cosmo-files were computed at the BP/def2-TZVPD-FINE//BP/def-TZVP level of theory (BP/def-TZVP for ELYTE and TZVP calculations).

Many of the studied cations have multiple conformers. At most 10 lowest chemical potential conformers were selected as input for the COSMO*therm* calculations. However, only conformers with chemical potentials within 2 kcal/mol of the lowest chemical potential were used, in order to avoid including conformers with low COSMO energies but high chemical potentials (Hyttinen, 2023). More specifically, COSMO*therm* gives high weights to conformers containing intramolecular H-bonds (Hyttinen and Prisle, 2020), because intramolecular H-bonds are favored in the COSMO energies (Kurtén et al., 2018). There is an excellent correlation between chemical potentials in water and in electrolyte solutions (see Fig. S6 of the Supplement). This indicates that conformer distributions are similar in pure water and electrolytes. The conformer sets for electrolyte calculations can therefore be selected using the chemical potentials of the ions in water.

## 3 Results and Discussion

### 3.1 Salt–Water Solutions

Water activities were calculated in solutions ranging from pure water to 0.2 mole fraction of salt. It should be noted that the water solubility of some salts at 298.15 K may be below 0.2 mole fraction. However, all salt was assumed to be dissolved in the solvent water as ions. Additionally, all cations in the calculations are singly charged for simpler comparison, even if some contain multiple amine groups. Figure 1 shows COSMO*therm*-derived water activities in all the studied salt solutions at 298.15 K. The values are given in Tables S1–S16 of the Supplement.







**Figure 1.** COSMO*therm*-derived water activities ($a_w$) in (a) sulfates, (b) bisulfates, (c) nitrates, (d) iodates and (e) methylsulfonates at 298.15 K. Activities were calculated in 0.01 intervals of salt mole fraction ($x_{\text{salt}}$) and plotted as lines for clarity.



Both the anion and cation affect the water activity of the solutions. Assuming that the ions investigated here follow the Hofmeister series, the order of cations should be similar regardless of counter ion. Based on COSMO*therm* estimates, the

order of the cations is fairly similar in sulfate, iodate and methylsulfonate salts. Similarly, bisulfate and nitrate salts have similar order for the cations. This indicates that the effect of the cation on water uptake also partially depends on the anion. Additionally, the order of the studied anions is not the same for all of the cations.

Some patterns can be seen in between the different functional groups of the cations. For example, water uptake is enhanced in the order: tertiary amine > secondary amine > primary amine. This order follows the hygroscopicity of aminium sulfate

salts (Rovelli et al., 2017) and is the same as Dawson et al. (2014) found for water uptake of aminium methylsulfonate salts (TMA > DMA > MMA). Additionally, two amine groups in the cation lead to lower water activity than one amine group, even though only one of the amine groups is protonated. This is seen in the comparison between TMA and TMEDA, as well as MMA and EDA.

Carboxylic acid groups have inconsistent effects on water activity. Amino acids have lower water activities than MMA

in some salts (bisulfate, nitrate) and higher in others (sulfate, iodate, methylsulfonate). Additionally, there is no clear trend between the studied amino acids with one (ALA, GLY, SER) or two (GLU, ASP) acid groups. On the other hand, water activity is lower in all five ARG (one carboxylic acid group) salts than in the corresponding AGM (no carboxylic acid group) salts.

In sulfate solutions, the water activities have the largest dependence on the cation compared to the other studied anions.

Additionally, GLY and GLU sulfates are predicted to inhibit water uptake at low salt concentrations (water activity coefficients above one). It should be noted that the ionic strength of the sulfate solutions is three times the ionic strength of the other studied solutions with the same salt mole fraction.

In their experiments, Sauerwein et al. (2015) observed similar osmotic coefficients of aqueous AM, MMA, DMA, TMA and DEA bisulfate solutions, indicating similar water activities independent of the cation. Similarly, the COSMO*therm*-derived

water activities in the bisulfate solutions are similar (within a factor of 1.1 in $x_{salt} = 0.1$ solutions) for all studied cations. Conversely, MMA, DMA, TMA and DEA sulfate solutions had lower experimental water activities than the corresponding bisulfate solutions, while water activities in AM bisulfate solutions were similar to those in AM sulfate solutions (Sauerwein et al., 2015). Similar trend is seen in the COSMO*therm*-derived water activities (Fig. 1).

Myllys (2023) studied the effect of water on the growth of acid–guanidine clusters in the gas phase computationally. In

the molecular level, water was seen to enhance particle formation at 298 K in systems containing nitric acid, as well as methylsulfonic acid at relatively low acid concentrations. No significant effect was seen in sulfuric acid systems. The same order in these three acids is seen in COSMO*therm* bulk-phase calculations. Nitrate enhances water uptake to the aqueous phase more than the other two anions, methylsulfonate enhances water uptake slightly less than nitrate, and the bisulfate has the weakest enhancing effect. However, COSMO*therm* predicts the lowest water activity in solutions containing sulfate

ions, indicating the strongest water uptake. However, since the concentration of guanidine was lower than the sulfuric acid concentration in the simulations of Myllys (2023), the sulfuric acid is expected to form bisulfate ions, not sulfate ions. If the correlation between COSMO*therm*-estimated water activities and the enhancement on cluster formation at high RH holds for





other acid-base pairs, the relative effect of RH on particle formation could be estimated using the computationally simple water activity calculations instead of the more time consuming molecular cluster calculations.

Higher equilibrium water content in aerosol particles indicates higher hygroscopic growth factor. Because the salts have different molar masses, mass fraction water content provides a more direct comparison for the hygroscopic factors of the salts. Table 1 shows the mass fraction water content of each of the studied salts at 70% RH as an example. Comparing the anions, the highest water content is found in either the sulfate of the nitrate salt for all studied cations. For most of the studied cations (all except TMEDA), the lowest water content in terms of mass fraction is in the iodate salt. COSMO*therm* predicted water mass

fractions at 70% RH vary between 0.19 (ASP-IO$_3$) and 0.64 (TMA$_2$-SO$_4$) for the studied salts.

**Table 1.** COSMO*therm*-predicted equilibrium water mass fractions ($m_w$) at 70% RH ($a_w = 0.7$) and 298.15 K.

|       | SO$_4{}^{2-}$ | HSO$_4{}^-$ | NO$_3{}^-$ | IO$_3{}^-$ | CH$_3$SO$_3{}^-$ |
|-------|------|------|------|------|------|
| AM    | 0.393 | 0.521 | 0.563 | 0.206 | 0.424 |
| MMA   | 0.454 | 0.454 | 0.551 | 0.328 | 0.486 |
| DMA   | 0.568 | 0.449 | 0.554 | 0.398 | 0.519 |
| TMA   | 0.636 | 0.462 | 0.562 | 0.448 | 0.544 |
| DEA   | 0.573 | 0.440 | 0.516 | 0.415 | 0.501 |
| EDA   | 0.448 | 0.396 | 0.494 | 0.343 | 0.461 |
| TMEDA | 0.551 | 0.404 | 0.487 | 0.416 | 0.482 |
| GUA   | 0.491 | 0.469 | 0.552 | 0.354 | 0.491 |
| AGM   | 0.410 | 0.343 | 0.413 | 0.321 | 0.394 |
| ARG   | 0.388 | 0.359 | 0.412 | 0.318 | 0.388 |
| ALA   | 0.303 | 0.409 | 0.457 | 0.257 | 0.381 |
| GLY   | 0.257 | 0.427 | 0.479 | 0.231 | 0.369 |
| SER   | 0.300 | 0.385 | 0.441 | 0.244 | 0.355 |
| ASP   | 0.229 | 0.383 | 0.419 | 0.191 | 0.311 |
| GLU   | 0.250 | 0.369 | 0.402 | 0.209 | 0.320 |
| HIS   | 0.329 | 0.306 | 0.371 | 0.290 | 0.352 |

## 3.2   Addition of water soluble organic

Next, an organic compound was added to the aqueous salt solutions to investigate the effect of the salts on water activity in the presence of an organic component. An isoprene-derived epoxydiol (*trans*-$\beta$-IEPOX; C$_5$H$_{10}$O$_3$) was selected as the organic compound, because of its high estimated gas-phase production rate in the atmosphere (100 Tg C per year; St. Clair et al.,

2016). Additionally, COSMO*therm* predicts miscibility between water and IEPOX and only slight deviation from ideality in all mixing ratios (Hyttinen et al., 2020). This is important, as limited solubility between water and the chosen organic would



lead to strong deviation from ideality for water by the organic. In such cases, liquid-liquid phase separation should be taken into account in the activity coefficient calculations.

### 3.2.1 Water content

Figure 2 shows the water activities in ternary salt–IEPOX–water solutions of three different methylsulfonate salts. The water activities were computed with 0.025 $x_w$ intervals and the $x_w$ values for each $a_w$ (0.99, 0.9, 0.8, 0.7, 0.6, 0.5, 0.4) were interpolated using the *spline* interpolation method of Matlab (MATLAB, 2019). The difference between interpolated $x_w$ values using 0.025 and 0.01 intervals was less than $10^{-6}$ mole fraction, indicating that *spline* finds a good fit to COSMO*therm*-estimated water activities calculated with 0.025 mole fraction intervals.

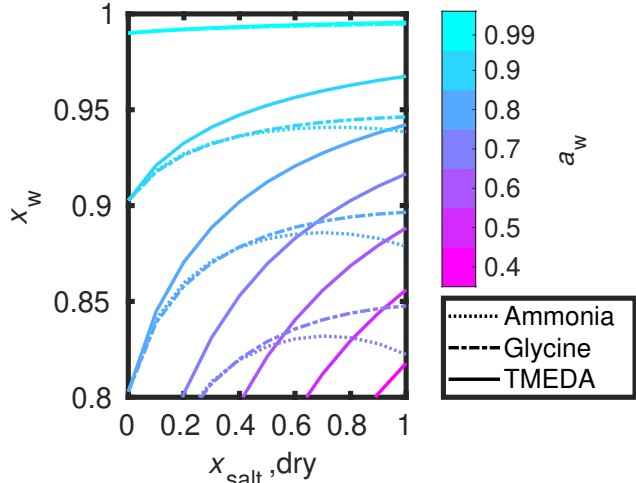

**Figure 2.** COSMO*therm*-derived water contents ($x_w$ in mole fraction) in aqueous solutions of three methylsulfonate salts and IEPOX. The values were calculated with 0.1 intervals of salt mole fraction in the dry mixture ($x_{salt}$,dry).

Ammonia, GLY and TMEDA have the weakest, the second weakest and the strongest effect on water uptake of the studied methylsulfonate salts, respectively (see Fig. 1e). Varying the mole ratio of salt and IEPOX shows a maximum equilibrium water mole fraction ($x_w$) in the AM methylsulfonate solutions at fixed RH (i.e., $a_w$). This indicates that adding an organic component may enhance water uptake compared to some pure salt particles. On the contrary, GLY and TMEDA methylsulfonate solutions reach their highest water content at a fixed RH in the particles containing no IEPOX.

Figure 3 shows a comparison between water contents (in mass fraction $m_w$) at equilibrium under 70% RH conditions ($a_w$=0.7) in pure salt (values in Table 1) and salt–IEPOX mixtures. Here, the salt:IEPOX ratio was kept constant at 1:1, corresponding to $x_{salt}$,dry $= 0.5$ in Fig. 2. Figures S7, S8 and S9 of the Supplement show similar comparisons at 90% RH, as well as of water mole fractions at 70% and 90% RH, respectively. Water activities in all studied salt–IEPOX mixtures with 1:1 mole ratio of salt and IEPOX, up to 0.1 mole fraction of salt, are shown in Fig. S10 of the Supplement.



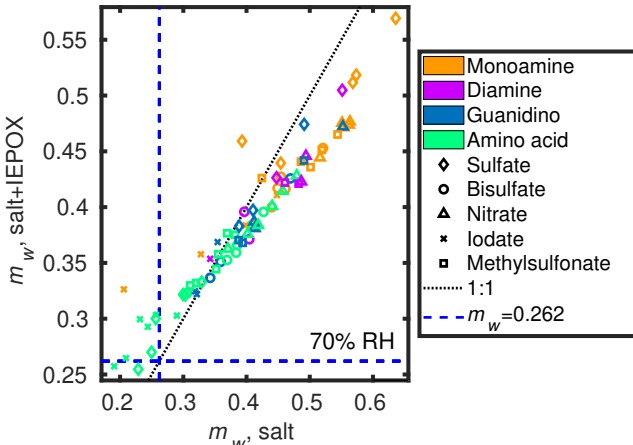

**Figure 3.** COSMO*therm*-derived equilibrium water contents ($m_w$ in mass fraction) in 70% RH at 298.15 K. The black dotted line shows the 1:1 correlation, the blue dashed lines are for $m_w = 0.262$, corresponding to water mass fraction in IEPOX at 70% RH.

The predicted equilibrium water content of particles containing small amines is the highest out of the studied amines, mainly due to their low molar mass compared to the polyamines. The equilibrium water content in IEPOX is predicted to be 0.699 mole fraction (0.262 mass fraction; dashed blue lines in Fig. 3) in 70% RH. The addition of salt to IEPOX with 1:1 mole ratio increases water uptake by 14 to 30 % in mole fraction depending on the salt. In mass fraction, the corresponding water uptakes increase by up to a factor of 2.2 (y-axis in Fig. 3). Pure salts have up to 36% higher water content in mole fraction, and 142% in mass fraction, than pure IEPOX in 70% RH (x-axis in Fig. 3).

### 3.2.2 Activity of the organic

The activities of IEPOX were computed in mixtures that contain equal mole fractions of IEPOX and salt. Overall, the IEPOX activities are higher in salt solutions than in pure water, indicating that all of the salts have a salting out effect on IEPOX. The solubility of IEPOX, and likely other hydrophilic organics, is reduced by aminium salts in the aerosol particles, compared to pure organic particles. Similar salting out behavior has been observed in COSMO*therm* calculations of ammonium sulfate and atmospherically relevant organics (Toivola et al., 2017; Hyttinen et al., 2020).

The order of IEPOX activities from the lowest to highest in the salt mixtures is bisulfate < nitrate < methylsulfonate < iodate < sulfate salts. The salting out effect of sulfate is much stronger than that of the other anions, likely due to the higher ionic strength. The order of the studied cations varies depending on both the counter ion and the salt mole fraction.

### 3.2.3 Organosulfate formation

Aoki et al. (2020) found that sulfate ions react with the epoxide group of IEPOX in the condensed phase to form an organosulfate species. However, the reaction between IEPOX and bisulfate was found to be slow, indicating that organosulfates are less likely to form in solutions containing equal mole fractions of sulfuric acid and amines (Aoki et al., 2020). Reactive uptake of





IEPOX into aqueous sulfate aerosols may counteract the strong salting out effect of sulfate salts for organic compounds that

can form ionic species by reacting with the sulfate ion.

Since organosulfates are likely to form from the selected organic (IEPOX), additional water activities were computed for mixtures containing equal mole fractions of aminium organosulfate salt and neutral amine. This mixture may arise if 1:1 mole ratio of sulfate salt and IEPOX react, neutralizing half of the aminium ions in the process:

$$\text{epoxide} + \text{SO}_4{}^{2-} + 2\text{RNH}_3{}^+ \rightarrow \text{organosulfate}^- + \text{RNH}_3{}^+ + \text{RNH}_2 \qquad \text{(R1)}$$

While the elemental content of the mixture remains the same, organosulfate formation reaction reduces the ionic strength of the mixture by neutralizing half of the amines and converting the doubly charged sulfate into singly charged organosulfate.

Water activities were additionally computed for organosulfate solutions that do not contain neutral amines. These mixtures correspond to either organosulfate formation from an epoxide reaction with bisulfate ion, or evaporation of the neutral amines after the organosulfate formation from a reaction with sulfate ion.

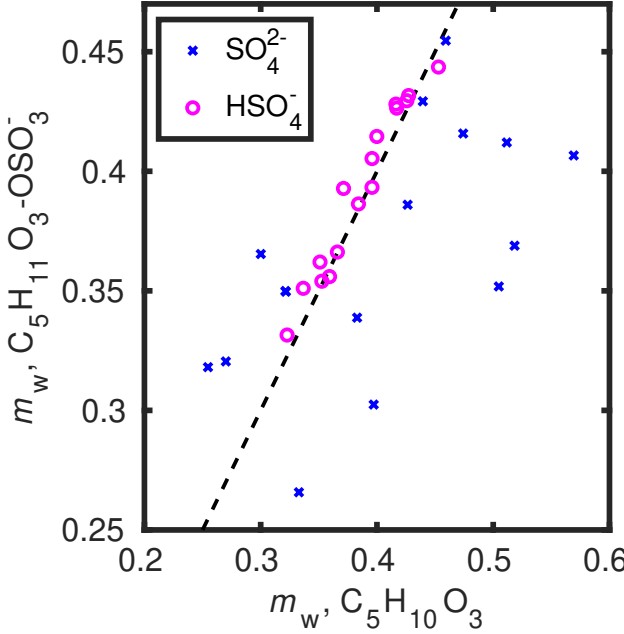

**Figure 4.** Water content ($m_w$ in mass fraction) in mixtures containing IEPOX (x-axis) and organosulfate (y-axis) under 70% RH conditions. 1:1 ratio is shown as the black dashed line.

Figure 4 shows the COSMO*therm*-derived water mass fraction in the bisulfate, sulfate (x-axis) and organosulfate (y-axis) mixtures at 70% RH. Similar water content is predicted for aminium bisulfate–IEPOX and aminium organosulfate mixtures (magenta circles in Fig. 4). On the other hand, larger differences are seen between aminium sulfate–IEPOX and the corresponding aminium organosulfate–amine mixtures. For most of the studied cations, water uptake is predicted to be stronger for the aminium sulfate–IEPOX mixture than for the corresponding aminium organosulfate–amine mixture. One reason may be





the difference in ionic strength between sulfate and organosulfate. Additionally, the COSMO*therm* estimates of sulfate salts may be less accurate than the of salts comprising singly charged non-spherical ions, such as bisulfate and organosulfate.

While amines can also react with epoxides in their neutral forms, they are likely to exist in protonated forms in acidic SOA. The nucleophilic strengths of protonated amines are much weaker than those of neutral amines (Stropoli and Elrod, 2015). This makes reactions between the studied aminiums and IEPOX unlikely. The nitrate ion can also act as a nucleophile and react

with the epoxy group. However, the product tertiary organonitrate is not as stable as the corresponding organosulfate, leading to water substituting the nitrate group to form a methyltetrol (Darer et al., 2011). The formation of an organonitrate species was therefore not considered here.

### 3.3 Addition of water-insoluble organic

A hydrophobic water-insoluble organic matter (WIOM; CC(=O)C1OC2OC(OC2(C)O1)C1=CC(C)=CC(C)=C1) proposed by

Kalberer et al. (2004) has often been used as a proxy in COSMO-RS calculations of atmospheric aerosol properties (Wania et al., 2014; Wang et al., 2017; Kurtén et al., 2016; Hyttinen et al., 2021; Lumiaro et al., 2021). COSMO*therm*-predicted solubility of water in WIOM is low (0.07 mole fraction; Hyttinen, 2023), indicating that relatively high RH is needed to increase the water content in pure WIOM particles. For example, at equilibrium in 70% and 90% RH, the predicted water mole fraction in WIOM is only 0.047 and 0.065, respectively. Adding a salt to the mixture increases equilibrium water content

compared to pure WIOM. However, high mole fractions of salt lead to the salting out of WIOM from the aqueous salt solution.

When the salt–WIOM–water mixtures phase separate into an aqueous phase and a organic–rich phase, the aqueous phase will contain mostly water and salt, while the organic–rich phase is almost pure WIOM. The COSMO*therm*-derived solubility of WIOM in water at 298.15 K is very low, only $2.5 \times 10^{-5}$ mole fraction. The solubility of WIOM further decreases with the addition of salt (salting out of WIOM). The predicted water activity in the aqueous phase is very close to that of the binary

salt–water mixtures (see Fig. S11 of the Supplement). The effect of very hygroscopic organic on water activity of salts is therefore expected to be negligible, as water activity in both phases at equilibrium is equal according to the LLE condition of Eq. (3).

### 4 Conclusions

Salts are expected to have a larger effect on water activity, and further equilibrium water content, than neutral organic com-

pounds. Additionally, the critical supersaturation is lowered through decreased water activity in salt containing particles. The water content of atmospheric aerosol also affects the aerosol radiative forcing, which is a crucial factor in climate models. It is therefore important to know how each ionic species in atmospheric aerosol affects water activity.

Water activities can be used to determine the hygroscopic growth and CCN activation of atmospheric aerosol particles. Knowledge on how different atmospheric ions affect water uptake and CCN activation of aerosol particles will help improve

climate models. COSMO*therm* calculations show how the different atmospheric anions and cations affect water activity. Most of the salts of this study enhance water uptake relative to ideal solutions (e.g., pure water).



Organosulfate and organic+bisulfate have similar effects the equilibrium water content. However, water uptake may be slower in organosulfate than in organic+sulfate mixtures. This indicates that the formation of organosulfate may reduce the water uptake ability compared to particles that do not contain organic compounds. This should be considered in models that use experimental hygroscopicities of pure sulfate salts to describe the hygroscopicities of mixed salt–organic particles. Future studies are needed to further investigate other ionic species and highly oxygenated multifunctional organics.

*Competing interests.* The author has declared that there are no competing interests.

*Acknowledgements.* I thank CSC - IT Center for Science, Finland, for computational resources.

*Financial support.* This project has received funding from the Academy of Finland, grant no 338171.



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
