# Peer review of "The effect of atmospherically relevant aminium salts on water uptake"

_EGUsphere, 2023_

## Referee Comment (RC2)

**General comments:**

I would like to thank the editor for inviting me to review this manuscript.

Hyttinen used the quantum chemistry based conductor-like screening model for real solvents (COSMO-RS) for estimating water activities atmospheric-relevant aminium salts as well as their mixtures with organic proxies. To the best of my knowledge, the author is the first to consistently apply density functional theory to determine the thermodynamic properties of atmospherically relevant aerosol-water systems. The data obtained are important for understanding the mechanism of aerosol particle formation and their hygroscopic and CCN properties. I recommend this manuscript to be published in ACP after the following issues to be addressed and modified.

Major comments:

- 1. Throughout the manuscript, I did not find error estimates of the modeled values. Is it possible to provide an estimate of the uncertainty of the computational results?
- 2. To assess the degree of non-ideality, I recommend that Figures S2-S5 show the water activity for an ideal solution as it was done in Fig.1.
- 3. It is difficult to analyze the calculation results presented in Fig.1. I recommend to split this data into several graphs.
- 4. As a potential user, I am concerned that different parameterization methods are used for different systems. Does this mean that additional parameterizations are required for systems with a different chemical composition? If so, it may be worthwhile to consider this issue, particularly to evaluate the prospect of using the DFT method to determine the thermodynamic properties of atmospherically relevant species as compared to other models that are easier to use.

Minor comments:

Line 35. Sulfate and bisulfate ions are not acids. Please correct.

Line 51 ... " $\mathbf{x}$  is the mixing state in mole fraction". Why not a ... mixing ratio.. or mole fraction of compound *i* ?

---

## Author Comment (AC1)

**Response to Referee #1**

Thank you for the insightful comments. Below are the referee comments in black, my responses to each point in red and changes made to the manuscript in blue.

This manuscript systematically investigated the water uptake potentials of a large library of the inorganic salts of amines with various functional groups (alkylamines, guanidinos, and amino acids), by computing the water activities of their aqueous solutions, as well as ternary mixtures of the salts, water and atmospheric-relevant organic proxies (either water soluble or water in-soluble). The author employed the Conductor-like Screening Model for Real Solvents (COSMO-RS) program with three parameterization methods (TZVP, FINE and ELYTE), depending on the specific cation in question. Based on the computational results, the water uptake potentials of amine salts depend more strongly on their cation chemical structures (eg, more substituents and more amino groups will enhance the water uptake of amine sulfates); but the anions may also affect such difference (eg, all bisulfate salts showed little difference in their aqueous solutions). Using an isoprene derived epoxydiol IEPOX as a proxy for water-soluble organics, the author showed that water soluble organics may enhance the water uptake of amine salts significantly, depending on the chemical structure of the salt. On the contrary, water in-soluble organics may simply become phase separated from the aqueous solutions to exert appreciable effects on the hygroscopicity of the salts.

While the manuscript has presented a thorough and comprehensive study on atmospheric-relevant amine salts in settings related to atmospheric chemical processes, I noticed a few items that may clarify some questions and further improve the structure and presentation of the manuscript:

1. Introduction. What are the (semi-)quantitative differences in water activity estimations using COSMOtherm program when compared with other methods (eg, the group attribution methods)? How about the differences in computational time, resource requirements, etc?

Author response: It would be interesting to compare water activities from COSMO*therm* and other models. Unfortunately it is not possible to calculate water activities in solutions containing aminium salts using group-contribution methods, because none of the group-contribution methods have been parametrized with experiments of aminium salts. This is why COSMO*therm* is the only option for computing properties of aminium salts. For example, the commonly used AIOMFAC contains only $NH_4^+$ out of the studied cations. See fig. R1 below for a comparison between AIOMFAC- and COSMO*therm*-derived water activities in aqueous ammonium salt solutions.

[Figure]

**Figure R1.** Comparison between COSMO*therm*- and AIOMFAC-derived water activities in the ammonium salts available in AIOMFAC.

Changes in manuscript (line 33): Though COSMO-RS calculations are much slower (several hours) than group-contribution methods ($\sim$ 1 second), mainly due to the quantum chemistry input needed for the COSMO-RS calculations, COSMO-RS is currently the only method capable of estimating water activities in aminium salts.

2. Methodology. Clearly it was difficult to find an "one-size-fit-all" computational method to address the complex research questions discussed here, and the manuscript states that FINE parameterization method was chosen for the amine salts and ELYTE for ammonium salts. However, when examining the Figs S1&S2, one could argue that the FINE may also be suitable for smaller amine salts. How was the "proper" parameterization method chose here? I think a more convincing approach is to calculate representative small amines using both FINE and ELYTE to see if the results are converging.

Author response: The ELYTE parametrization was selected for only ammonium sulfate, not all ammonium salts. This was mentioned in line 78 of the manuscript "The FINE parametrization is therefore used for all of the studied salts, with the exception of ammonium sulfate, which is calculated using the ELYTE parametrization." The ELYTE parametrization is only meant for calculations involving ionic species and should not be used for charge neutral systems. The FINE parametrization uses the higher level of theory (DFT) input, while TZVP and ELYTE use the lower level of theory input. For both neutral and ionic systems, the two levels of theory (BP/TZVP and BP/TZVPD) lead to different water activities, though the differences

between the parametrizations are larger in ionic systems than in neutral systems. Generally, the FINE parametrization should lead to more accurate results than the lower level TZVP parametrization and that is why it was chosen for the calculations here. These points were clarified in the manuscript.

Changes in manuscript (line 73): Based on these recommendations, the use of ELYTE should be limited to systems for which it was developed, unless the calculated values can be corroborated by experiments.

(line 78): Additionally, the current electrolyte parametrization is only available for the lower level of theory BP/TZVP cosmo files and not for the higher level of theory BP/TZVPD cosmo files, corresponding to the FINE parametrization. In case there are systematic errors within the different parametrizations or levels of theory, the comparison of the different salts is more reliable when the same parametrization is used in all calculations.

3. Results&Discussion. The point above also brings up another major question: what is the estimated uncertainty in the computational results presented in the manuscript? Some of the traces in the figures were quite close to each other and the lack of error bars was making it difficult to judge if the difference was within the uncertainties of the methods.

Author response: Thank you for the question. It is difficult to quantify the uncertainty of COSMO*therm* calculations, because the performance of the model depends on the compound.

Changes in manuscript (line 126): Due to uncertainties in the calculated values, it is possible that there are no real differences between the water activities of the different aminium bisulfate solutions.

(line 102): Unfortunately, determining reliable error estimates of water activities in these salts is not possible due to lack of experimental data of similar salts. However, the errors are likely to be systematic for similar compounds, such as aminium ions. The order of the cations may therefore be more reliable than the order of the anions.

4. Results&Discussion. This study was very comprehensive with a lot of results, which leads to, in my opinion, very dense figures that are difficult to read and comprehend. I would like to recommend major reorganizations of the results for better presentations and support for the flow of the discussion. For example, some panels of Fig. 1 can go into supplementary, but others can be split into less busy ones (eg, Fig.1a may be divided into three panels, one for each amine group to highlight the differences).

Author response: Thank you for the suggestion. I have made a new figure that better highlights the differences between both anions and cations. The original Figure 1 was moved to the Supplement.

Changes in manuscript (line 101): Figure 1 shows COSMO*therm*-derived water activities in selected salt solutions at 298.15 K.

[Figure]

**Figure 1.** COSMO*therm*-derived water activities ($a_w$) in aqueous (a) amine, (b) guanidino and (c) amino acid bisulfate solutions, as well as selected (d) sulfate, (e) methylsulfonate and (f) nitrate solutions at 298.15 K. Activities were calculated in 0.01 intervals of salt mole fraction ($x_{\text{salt}}$) and plotted as lines for clarity. The black line represents ideality $a_w = x_w$.

---

## Author Comment (AC2)

**Response to Referee #2**

Thank you for the insightful comments. Below are the referee comments in black, my responses to each point in red and changes made to the manuscript in blue.

Hyttinen used the quantum chemistry based conductor-like screening model for real solvents (COSMO-RS) for estimating water activities atmospheric-relevant aminium salts as well as their mixtures with organic proxies. To the best of my knowledge, the author is the first to consistently apply density functional theory to determine the thermodynamic properties of atmospherically relevant aerosol-water systems. The data obtained are important for understanding the mechanism of aerosol particle formation and their hygroscopic and CCN properties. I recommend this manuscript to be published in ACP after the following issues to be addressed and modified.

Major comments:

1. Throughout the manuscript, I did not find error estimates of the modeled values. Is it possible to provide an estimate of the uncertainty of the computational results?

Author response: Thank you for this important question. See the response to comment 3 from referee 1.

2. To assess the degree of non-ideality, I recommend that Figures S2-S5 show the water activity for an ideal solution as it was done in Fig.1.

Thank you for this suggestion. The ideality lines have been added to Figures S1-S5 in the Supplement.

3. It is difficult to analyze the calculation results presented in Fig.1. I recommend to split this data into several graphs.

Author response: Thank you for the suggestion. I have made a new figure that better highlights the differences between both anions and cations. See the response to comment 4 from referee 1.

4. As a potential user, I am concerned that different parameterization methods are used for different systems. Does this mean that additional parameterizations are required for systems with a different chemical composition? If so, it may be worthwhile to consider this issue, particularly to evaluate the prospect of using the DFT method to determine the thermodynamic properties of atmospherically relevant species as compared to other models that are easier to use.

Author response: I fully agree with this. Ideally, one could use the FINE parametrization for all calculations in the future, if the electrolyte model is extended to the BP/TZVPD level of theory. The ELYTE parametrization was used here for ammonium sulfate only because the clear disagreement between experiments and the FINE parametrization. The parametrizations of COSMO-RS programs are generally based on the level of theory used to obtain the quantum chemistry input files, so that only one parametrization exists for one level of theory and the same parametrization should be used for all compounds. One exception are systems containing small ions. For this reason, the developers of COSMO*therm* have added an electrolyte parametrization for one of the levels of theory. The choice between levels of theory is usually based on time constrains, since the BP/TZVPD calculations are significantly more time consuming than the BP/TZVP calculations. For larger molecules, e.g.

proteins, using the BP/TZVPD level of theory may not be feasible. Otherwise, the COSMO*therm* developers recommend using the BP_TZVPD_FINE_21 parametrization (BIOVIA COSMO*therm*, 2021). See also my response to comment 1 from referee 1.

Minor comments:

35     Line 35. Sulfate and bisulfate ions are not acids. Please correct.

Author response: Thank you for noticing this. The sentence has been changed in the manuscript.

Changes in manuscript (line 35): Sulfuric acid ($H_2SO_4$) is derived from anthropogenic emissions of $SO_2$, nitric acid ($HNO_3$) is produced from both anthropogenic and natural processes, and iodic acid ($HIO_3$) and methylsulfonic acid ($CH_3SO_3H$) are more abundant in marine environments. In aqueous solutions, these strong acids are deprotonated to form sulfate ($SO_4^{2-}$) and

40   bisulfate ($HSO_4^-$), nitrate ($NO_3^-$), iodate ($IO_3^-$) and methylsulfonate ($CH_3SO_3^-$), respectively.

Line 51 ... "**x** is the mixing state in mole fraction". Why not a ... mixing ratio.. or mole fraction of compound i ?

Author response: **x** contains the mole fractions of all compounds in the system, not just the mole fraction of compound i. To avoid confusion, the wording was changed.

Changes in manuscript (line 51): where **x** is the composition of the mixture, ...

**45 References**

BIOVIA COSMO*therm*: Reference Manual, 2021.

---

## Author Response (AR2)

**Response to the Editor**

Thank you for the additional comments. Below are the editor comments in black, my responses to each point in red and changes made to the manuscript in blue.

I would like to thank the author for their response and revision, which improved the manuscript. Before the manuscript can be published in ACP, I would like to reiterate on two points brought up by the reviewers.

5 1 - Comparison with AIOMFAC (Reviewer 1, Comment 1). The supplied graphic R1 is very useful for the reader to gauge the overall uncertainty/variability between different methods. I suggest to include such a graphic into the electronic supplement along with a qualifying sentence in the main manuscript.

Additional question: What determines the aw range for calculations (ends of the lines in R1), is this  $x_{salt} = 1$ ? It would be helpful if this could be indicated.

10 Author response: I have added the figure to the Supplement with a description of the comparison in the main text. The water activities in Fig. R1 (now Fig. S1) were calculated for salt mole fractions between 0 and 0.2, corresponding to all other calculations in the paper. This was added to the caption of the figure.

Changes in manuscript (line 33): COSMO-RS and AIOMFAC predict similar water activities in  $(NH_4)_2SO_4$ ,  $NH_4HSO_4$ ,  $NH_4NO_3$  and  $NH_4IO_3$ , within a factor of 1.08 and 1.30 in aqueous solutions with 0.1 and 0.2 mole fraction of the salt, respectively (see Fig. S1 of the Supplement).

Changes in Supplement:

15

Figure S1: Comparison between COSMO*therm*- and AIOMFAC (AIOMFAC-web, 2023)-derived water activities in ammonium salts at 298.15 K. Water activities were calculated for solutions with salt mole fractions from 0 to 0.2. The black dotted line shows the 1:1 line.

- 20 2 Both reviewers asked for an estimate of model uncertainty (Reviewer 1, Comment 3; Reviewer 2, Comment 1). While a good point was made by the author that there are no measurements for comparison, the reader would still massively benefit from a general estimation. For example, it would be interesting to give such a comparison for species for which data are available (e.g. including data into a figure complementing Fig. R1), together with a statement on whether similar uncertainties could be expected for the aminium species in question.
- I am aware that this is not the main focus of this manuscript, which are aminium salts, so also a reproduction of or simple reference to existing literature could be helpful here.

Author response: I have added calculated error estimates based on the existing ammonium and aminium salt experiments of Figs S1–S3. This should provide the most relevant error estimate for the aminium salts.

Changes in manuscript (line 79: Using the selected parametrizations, COSMO*therm* agrees with the experimental water 30 activities of aminium sulfate, bisulfate and nitrate solutions ( $x_{salt} < 0.2$ ) within factors of 1.39, 1.18 and 1.23, respectively. Similar uncertainties can be expected for other aminium salts. Additionally, as can be seen from Figs S2–S4 of the Supplement, the disagreement between calculations and experiments increases with the increasing salt mole fraction in the solution. Furthermore, I have small remark:

3 - 1. 73 - "The highest level of theory parametrization in COSMOtherm (BP\_TZVPD\_FINE\_21; abbreviated FINE; using
BP/TZVPD//BP/TZVP level cosmo files) works poorly with the strong charges of small ions, such as atomic ions and strong

small semi-spherical ions (such as SO42-, NH4+)." - Can you clarify the literature reference on this statement and other statements in this paragraph?

Author response: Thank you for this clarification, I have added the reference BIOVIA COSMOtherm (2021) to all of the sentences instead of the end of the paragraph.

40

After these minor comments are addressed, I would be happy to accept the manuscript for publication. Thank you very much and best regards, Thomas Berkemeier

**References**

AIOMFAC-web: version 3.05, http://www.aiomfac.caltech.edu, last access: 28 August 2023, 2023.

BIOVIA COSMOtherm: Reference Manual, 2021.